# Linear relationship between effective radius and precipitation water content near the top of convective clouds: measurement results from ACRIDICON-CHUVA campaign

Ramon Campos Braga[1], Daniel Rosenfeld[2], Ovid O. Krüger[1], Barbara Ervens[3], Bruna A. Holanda[1], Manfred Wendisch[4], Trismono Krisna[4], Ulrich Pöschl[1], Meinrat O. Andreae[1,5,6], Christiane Voigt[7,8], and Mira L. Pöhlker[1]

[1]Multiphase Chemistry Department, Max Planck Institute for Chemistry, 55128 Mainz, Germany.
[2]Institute of Earth Sciences, The Hebrew University of Jerusalem, Jerusalem, Israel.
[3]Université Clermont Auvergne, CNRS, SIGMA Clermont, Institut de Chimie de Clermont-Ferrand, 63000 Clermont-Ferrand, France.
[4]Leipziger Institut für Meteorologie (LIM), Universität Leipzig, Stephanstr. 3, 04103 Leipzig, Germany.
[5]Scripps Institution of Oceanography, University of California San Diego, La Jolla, CA 92037, USA
[6]Department of Geology and Geophysics, King Saud University, Riyadh, Saudi Arabia
[7]Institute of Atmospheric Physics, German Aerospace Center (DLR), 82234 Oberpfaffenhofen, Germany
[8]Johannes Gutenberg University Mainz, 55099 Mainz, Germany

**Correspondence:** Ramon Campos Braga (r.braga@mpic.de) and Mira L. Pöhlker (m.pohlker@mpic.de)

**Abstract.** Quantifying the precipitation within clouds is a crucial challenge to improve our current understanding of the Earth's hydrological cycle. We have investigated the relationship between the effective radius of droplets and ice particles ($r_e$) and precipitation water content ($PWC$) measured by cloud probes near the top of growing convective cumuli. The data for this study were collected during the ACRIDICON-CHUVA campaign on the HALO research aircraft in clean and polluted conditions over the Amazon Basin and over the western tropical Atlantic in September 2014. Our results indicate a threshold of $r_e \sim 13\ \mu$m for warm rain initiation in convective clouds, which is in agreement with previous studies. In clouds over the Atlantic Ocean, warm rain starts at smaller $r_e$, likely linked to the enhancement of coalescence of drops formed on giant cloud condensation nuclei. In cloud passes where precipitation starts as ice hydrometeors, the threshold of $r_e$ is also shifted to values smaller than 13 $\mu$m when coalescence processes are suppressed and precipitating particles are formed by accretion. We found a statistically significant linear relationship between $PWC$ and $r_e$ for measurements at cloud tops, with a correlation coefficient of $\sim 0.94$. The tight relationship between $r_e$ and $PWC$ was established only when particles with sizes large enough to precipitate (drizzle and raindrops) are included in calculating $r_e$. Our results emphasize for the first time that $r_e$ is a key parameter to determine both initiation and amount of precipitation at the top of convective clouds.

# 1 Introduction

Convective cloud formation and precipitation processes have different characteristics depending on the atmospheric thermodynamic conditions and aerosol particle concentration (Reutter et al., 2009; Rosenfeld et al., 2008; Tao et al., 2012). In clean air masses, low concentrations of cloud condensation nuclei (CCN) lead to clouds with relatively fewer droplets ($\sim$50 – 200 cm$^{-3}$) at cloud base but with larger sizes (Twomey, 1974; Andreae et al., 2004; Rosenfeld et al., 2008; Braga et al., 2017a; Sorooshian et al., 2019). These droplets initially grow fast by condensation and subsequently coalesce rapidly into raindrops. In polluted air masses, high concentrations of CCN produce clouds with high concentrations of small drops at cloud base, which can exceed 1000 cm$^{-3}$. The small and numerous drops grow slowly by condensation due to the high competition for water vapor. In such a case, the coalescence of cloud drops into raindrops is suppressed, and thus, raindrop formation takes place from the melting of ice particles (Andreae et al., 2004; Braga et al., 2017b; Khain et al., 2008; Rosenfeld et al., 2008; Berg et al., 2008).

Over the Tropics, convective clouds and mesoscale convective systems account for most of precipitation and severe weather (Liu et al., 2007; Roca et al., 2014; Zipser et al., 2006). In the Amazon Basin, the formation and development of precipitation-forming processes of convective clouds occur at different levels of atmospheric pollution (Andreae et al., 2004; Pöhlker et al., 2016). Previous studies (such as Roberts et al. (2001); Martin et al. (2016)) have shown that during the wet season (Feb-May), low concentrations of CCN particles, mainly consisting of forest biogenic aerosols, are found in the Amazon Basin ($\sim$ 200-300 cm$^{-3}$ for 1% of supersaturation), leading to the formation of shallow convective clouds with low ice water content and lightning activity (Albrecht et al., 2011; Williams et al., 2002). These characteristics of the Amazonian clouds and CCN concentrations during the wet season led several authors to refer this region as a "green ocean" to highlight its similarity with maritime-like regions (e.g., Pöhlker et al. (2016); Roberts et al. (2001); Martin et al. (2016)). On the other hand, during the dry season(Aug-Oct), the background concentrations of CCN over the Amazon can reach values $\sim$ 10 times higher than those of the green ocean (Artaxo et al., 2002, 2013; Pöhlker et al., 2018; Roberts et al., 2003). The increase of particle concentrations results from forest, savanna, and agricultural fires that release large amounts of biomass burning aerosols over the pristine rain forest (Andreae et al., 1988). Such conditions inhibit the formation of shallow precipitating clouds and invigorate the ice processes within convective clouds and their lightning activity (Albrecht et al., 2011; Williams et al., 2002; Rosenfeld et al., 2008).

Braga et al. (2017b) have described the general characteristics of growing convective cumulus formed over the Amazon basin and Atlantic Ocean based on in situ measurements. The measurements were performed during the ACRIDICON-CHUVA (Aerosol, Cloud, Precipitation, and Radiation Interactions and Dynamics of Convective Cloud Systems–Cloud Processes of the Main Precipitation Systems in Brazil: A Contribution to Cloud Resolving Modeling and to the Global Precipitation measurements) campaign in the Amazonian dry season in September 2014 (Wendisch et al., 2016). During the campaign, cloud profiling flights were performed in regions of different pollution levels and thermodynamic conditions. Braga et al. (2017b) showed that the heights of cloud base are higher over the continental Amazon due to the smaller relative humidity in comparison with the maritime region. Convective clouds formed over the Atlantic Ocean near the Brazilian coast have smaller cloud droplet concentrations at cloud base due to the smaller concentration of aerosol and updraft velocities below cloud base. For convective clouds formed over forested and deforested regions, larger aerosol concentration and updrafts were observed below

cloud base, leading to larger droplets activated at cloud base. The precipitating particles were formed mostly by coalescence of drops at temperatures above 0˚C over ocean. Over the forest, lightly polluted air masses were found and the precipitation

initiation (liquid raindrops) was observed near 0˚C. For very polluted air masses, found over the deforestation arc region, the collision and coalescence processes were totally suppressed and the formation of precipitating particles took place at higher altitudes as ice hydrometeors. In these cases, precipitating particles were formed mostly by accretion processes at temperatures below 0˚C, when the growth of ice hydrometeors took place from collision with supercooled drops that freeze completely or partially upon contact.

The relationship between particle sizes and precipitation is associated with the coalescence rate which increases in direct proportion to the cloud droplet effective radius ($r_{ec}^5$) (Freud and Rosenfeld, 2012). Previous studies have found $r_{ec}$ between 13 $\mu$m and 14 $\mu$m as a suitable threshold for precipitation initiation (Freud and Rosenfeld, 2012; Rosenfeld and Gutman, 1994; Braga et al., 2017b). The relation between rain initiation and $r_{ec}$ is associated with the increase of both the drop-swept volume and collision efficiency. The collision efficiency of drops increases as a function of their sizes (Khain and Pinsky, 2018). For

raindrops, this value is close to unity, and is several times larger than that for small drops ($r < 10$ $\mu$m) (Pinsky et al., 2012). The collision and coalescence processes of liquid drops and the accretions processes at supercooled temperatures have strong effect on the broadening of the particle size distribution and thus particle sizes. In this study, we have investigated measurements of the effective radius ($r_e$) of cloud particles and the rain and ice precipitation water content (*PWC*) using data from cloud probes, measured at the cloud tops of growing convective cumulus during the ACRIDICON-CHUVA campaign. We focus our analysis

on flights in which precipitation was found in the cloud tops of convective clouds. Our findings shown in the next sections describe the tight relationship between $r_e$ and *PWC* for in situ measurements in cloud tops in different pollution states and temperature levels. We show that $r_e$ determines both the initiation and amount of precipitation at the top of convective clouds.

## 2    Methods

### 2.1    Data and Instrumentation

#### 2.1.1    Research flights

The data used in this study are droplet and ice particle concentrations measured in convective clouds by cloud probes mounted on the HALO aircraft during the ACRIDICON–CHUVA campaign (Wendisch et al., 2016). The HALO aircraft was equipped with a meteorological sensor system (BAsic HALO Measurement And Sensor System - BAHAMAS) located at the nose of the aircraft (Wendisch et al., 2016). The description of typical meteorological measurements can be found in Mallaun et al.

(2015). The HALO flights took place over the Amazon region under various conditions of aerosol concentrations and land cover. Figure 1 shows the flight tracks of cloud profiling flights where precipitation was observed in convective clouds (Braga et al., 2017b). The region of measurements is indicated by circles for each flight. Convective clouds formed in clean air masses were found above the Atlantic Ocean during flight AC19 (in blue, Fig.1). Flights AC09 and AC18 took place in lightly polluted

conditions over the tropical rain forest (in green, Fig.1). Clouds forming in deforested regions in very polluted (biomass burning) environments were measured during flights AC07 and AC13 (in red and orange, Fig.1).


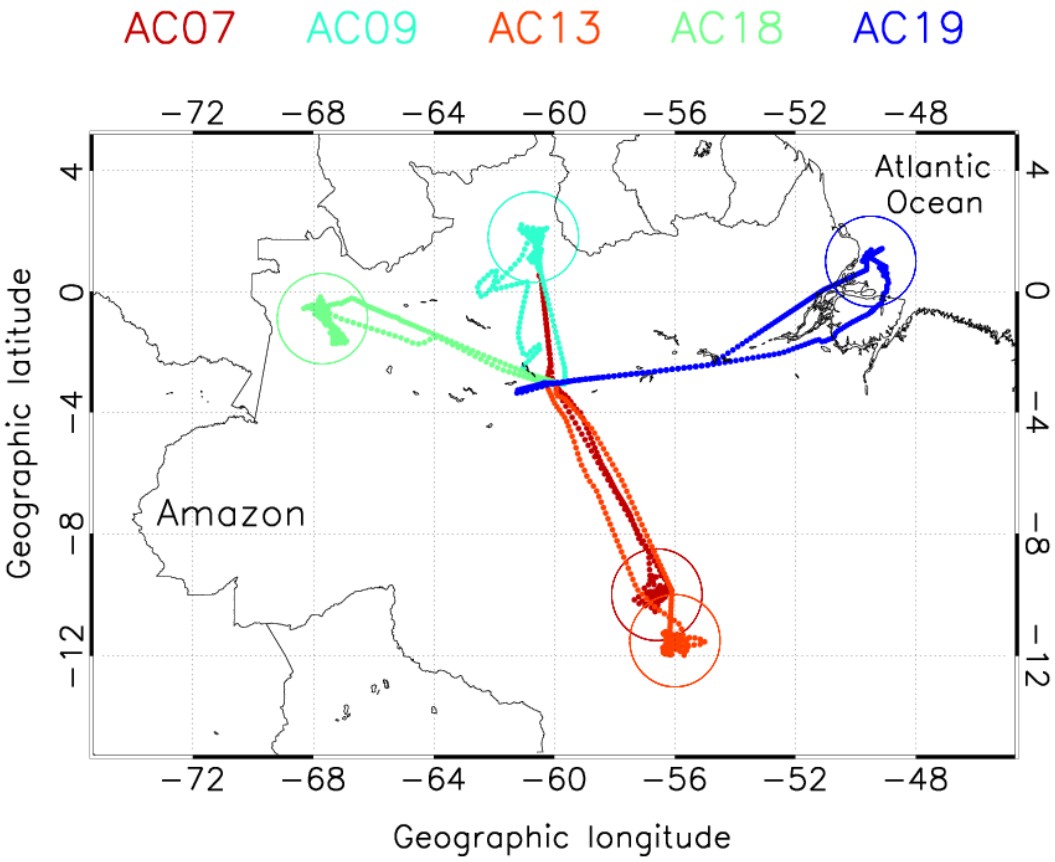

**Figure 1.** HALO flight tracks during the ACRIDICON–CHUVA experiment. The flight number is indicated at the top by colors. Colored circles indicate the region of cloud profiling in each flight. Convective clouds formed in clean air masses were found above the Atlantic Ocean during flight AC19 (shown in blue). Clouds in lightly pollution conditions were found during Flights AC09 and AC18 over the tropical rain forest (shown in green). Polluted clouds forming in deforested regions were measured during flights AC07 and AC13 (shown in red and orange, respectively). The average aerosol particle concentration measured near cloud bases during flights AC07, AC09, AC13, AC18, and AC19 were 2498 cm$^{-3}$, 821 cm$^{-3}$, 4093 cm$^{-3}$, 744 cm$^{-3}$ and 465 cm$^{-3}$, respectively (Cecchini et al., 2017).

### 2.1.2   Cloud particle measurements

Cloud particle number concentrations and size distributions were measured by the Cloud Combination Probe (CCP) mounted on board the HALO. The CCP combines two detectors, the Cloud Droplet Probe (CDP) and the grayscale Cloud Imaging Probe (CIPgs). The CDP is an open-path instrument that detects forward-scattered laser light from cloud particles as they pass

through the detection area (Lance et al., 2010). The CIP records 2-D shadow-cast images of cloud elements. The identification of water drops and ice hydrometeors were performed by Braga et al. (2017b) from the occurrence of visually spherical and non-spherical shapes of the shadows. The combination of CCP–CDP and CCP–CIPgs information provides the ability to measure particles within clouds for nearly the same air sample volume.

Cloud particle size distributions (DSDs) between 3 and 960 $\mu$m in diameter were measured at a temporal resolution of 1 s by
the CCP-CDP and CCP–CIPgs (Brenguier et al., 2013; Weigel et al., 2016). Each DSD spectrum represents 1 s of flight path (covering between 63 m and 112 m of horizontal distance at the aircraft speed). Details about the cloud probe measurements characteristics during ACRIDICON-CHUVA campaign are described in Wendisch et al. (2016), Weigel et al. (2016), Braga et al. (2017a) and Braga et al. (2017b). In this study, a cloud pass is assumed when the total water content (*TWC*) exceeds 0.05 g m$^{-3}$ and the number concentration of drops (*N_d*) exceeds 20 cm$^{-3}$. This criterion was applied to avoid cloud passes well mixed
with subsaturated environment air (RH < 100%) and counts of haze particles, typically found at cloud edges and dissipating convective clouds during the ACRIDICON-CHUVA campaign (Braga et al., 2017b). The $N_d$ and *TWC* are defined as:

$$N_d = \int_{1.5\mu m}^{480\mu m} N(r)dr \tag{1}$$

and

$$TWC = \frac{4\pi}{3}\rho \int_{1.5\mu m}^{480\mu m} r^3 N(r)dr \tag{2}$$

where $N$ is the particle number concentration (cm$^{-3}$), $\rho$ is the particle density, and $r$ the particle radius ($\mu$m).

## 2.2 Analysis of cloud properties

Previous studies (e.g., Freud and Rosenfeld (2012); Braga et al. (2017b)) have calculated $r_e$ using data of particle number concentration with radii between 1.5 $\mu$m and 25 $\mu$m ($r_{ec}$), which does not include precipitating particles. Here, the relationship between cloud particle sizes and *PWC* is investigated by calculating $r_e$ taking into account the concentration of particles with
precipitating sizes (1.5 $\mu$m < $r$ ≤ 480 $\mu$m). Precipitating particles are considered those with terminal fall speeds large enough ( > ∼ 0.5 m s$^{-1}$) to survive evaporative dissipation over a distance of the order of several hundred meters (Beard, 1976). Droplets smaller than drizzle particles fall slowly enough from most clouds that they evaporate before reaching the ground. The size range of the *PWC* calculation includes particles with drizzle (25 $\mu$m ≤ $r$ ≤ 125 $\mu$m) and raindrop (125 $\mu$m < $r$ ≤ 480 $\mu$m) sizes. The drizzle water content (*DWC*) and *PWC* are calculated using the size range of drizzle and raindrop in Eq.
2. Precipitating particles in this size range ($r$ > 25 $\mu$m) were often imaged by cloud probes within convective cumulus during ACRIDICON-CHUVA campaign (Braga et al., 2017a).

Here, we performed our analysis along the following general steps.

a. The relationship between the measured $r_e$ and *PWC* near the top of convective clouds is calculated based on CCP measurements (described in Sect. 3.1).

b. The precipitation probability as a function of the measured $r_e$ and *DWC* near the top of convective clouds is detailed in Sect. 3.2.

c. The vertical development of cloud particles growth near the top of growing convective cumuli is described for clean and polluted conditions in Sect. 3.3.

d. The extent of agreement between $r_e$ and *PWC* measured near cloud tops is discussed in Sect. 4 and 5.

To this end, the following cloud properties were taken into account during our analysis.

Effective radius ($r_e - [\mu\mathrm{m}]$):

$$r_e = \frac{\int_{1.5\mu m}^{480\mu m} r^3 N(r)\, dr}{\int_{1.5\mu m}^{480\mu m} r^2 N(r)\, dr} \tag{3}$$

Cloud particle effective radius ($r_{ec} - [\mu\mathrm{m}]$):

$$r_{ec} = \frac{\int_{1.5\mu m}^{25\mu m} r^3 N(r)\, dr}{\int_{1.5\mu m}^{25\mu m} r^2 N(r)\, dr} \tag{4}$$

Mean radius ($r_M - [\mu\mathrm{m}]$):

$$r_M = \frac{1}{N} \int\limits_{1.5\mu m}^{480\mu m} r\, N(r)\, dr \tag{5}$$

Mean volume radius ($r_V - [\mu\mathrm{m}]$):

$$r_V = \left( \frac{\int_{1.5\mu m}^{480\mu m} r^3 N(r)\, dr}{\int_{1.5\mu m}^{480\mu m} N(r)\, dr} \right)^{\frac{1}{3}} \tag{6}$$

Modal radius ($r_{MOD} - [\mu\mathrm{m}]$) is the radius in which:

$$\left. \frac{\partial N(r)}{\partial r} \right|_{1.5\mu m}^{480\mu m} = 0 \tag{7}$$

Cloud mass ratio (*CMR*):

$$CMR = \frac{\int_{1.5\mu m}^{25\mu m} r^3 N(r) dr}{\int_{1.5\mu m}^{480\mu m} r^3 N(r) dr} \tag{8}$$

Precipitation mass ratio (*PMR*):

$$PMR = \frac{\int_{25\mu m}^{480\mu m} r^3 N(r) dr}{\int_{1.5\mu m}^{480\mu m} r^3 N(r) dr} \tag{9}$$


The uncertainties of the calculated values of $r_e$, $r_{ec}$, $r_V$, $r_{MOD}$, *CMR*, and *PMR* are $\sim$ 10 % (Braga et al., 2017a, b). The uncertainties of calculated *DWC* and *PWC* are $\sim$ 30 %. Furthermore, Braga et al. (2017b) showed that water drops were observed near cloud tops for air temperatures (*T*) warmer than -9°C over the Amazon basin. For $T \leq$ - 9°C, ice initiation was found. Therefore, for cloud particles measured at $T >$ -9°C the density of water (1 g cm$^{-3}$) is used in calculations of cloud

properties. For $T \leq$ - 9°C, Braga et al. (2017b) showed that mostly graupel and frozen drops were imaged by the CIPgs, and thus we assume in our calculations that the density of frozen particles is 0.9 g cm$^{-3}$ ($\sim$ the density of pure ice). Results assuming smaller density for ice particles are also shown in the supplement. In addition, we assume a spherical shape for water and ice particles in our calculations. The density of ice particles within clouds is associated with the microphysical mechanism of their growth. An ice particle originated from a frozen drop or ice crystal due to accretion processes to an irregular or roundish particle

has bulk density of 0.8 g cm$^{-3}$ < $\rho$ < 0.99 g cm$^{-3}$ (Pruppacher and Klett, 1997). Furthermore, the density of rimed ice particles has a strong influence on the denseness of packing of the cloud drops frozen onto the ice crystal. These factors can result in graupel particles with densities ranging between 0.05 g cm$^{-3}$ and 0.9 g cm$^{-3}$. Ice particles formed by deposition of water vapor and collision of snow crystals (e.g., snow-flakes, ice crystals, needles, columns, and sheets) typically have low densities ($\sim$ 0.05 g cm$^{-3}$ < $\rho$ < 0.5 g cm$^{-3}$). Therefore, based on the type of particles imaged by the CIPgs during our measurements we

assume that the uncertainty in the calculated $r_e$ and *PWC* is small in comparison to the measurement uncertainty (i.e., $\sim$10% and $\sim$30% for $r_e$ and *PWC*, respectively).

## 3   Results

### 3.1   Comparison of measured $r_e$ and *PWC* near the top of convective clouds

Figure 2 shows the measured $r_e$ and *PWC* near the top of convective clouds. The precipitation was found in liquid and solid

phases for temperatures ranging between -26°C and 10°C (see Fig. S1 for $T$-$r_e$ profiles). The relationship between $r_e$ and *PWC* can be well expressed by a linear function (R$^2$ $\sim$ 0.89) for liquid and frozen precipitation. The high correlation between $r_e$

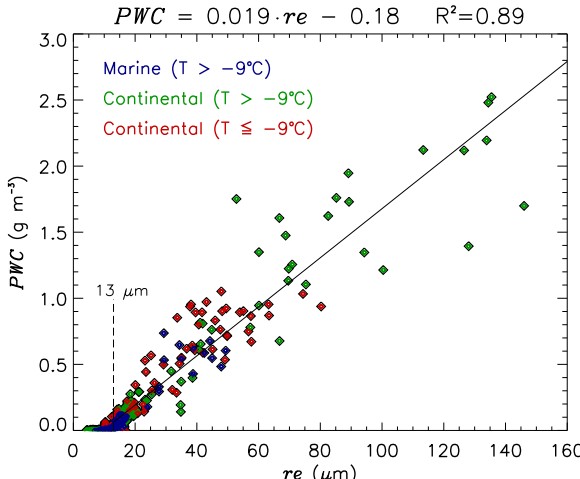

**Figure 2.** Effective radius of cloud particles ($r_e$) vs. precipitation water content (*PWC*) measured near cloud top of convective clouds over the Atlantic Ocean for temperatures (*T*) warmer than -9 ˚C (in blue), and over the continent for T > -9 ˚C (in green) and T ≤ -9 ˚C (in red). The black line indicates the fit of *PWC* as a function of $r_e$ (shown on the top of the graphic). The $r_e$ threshold of 13 $\mu$m applied for the fit is based on the value of $r_e$ at which light precipitation starts (drizzle water content > 0.01 g m$^{-3}$). The coefficient of determination ($R^2$) from the fit function of this analysis is shown on the top.

and *PWC* was not found for $r_e$ when considering only the cloud drop size range ($r < 25$ $\mu$m) [see Fig. S2]. When precipitating particles are neglected in the calculation of $r_e$, the large increase of precipitation mass is not captured, and thus, $r_{ec}$ values do not exceed 17 $\mu$m in our analysis. Nevertheless, these characteristics do not prevent $r_{ec}$ from identifying the threshold of
precipitation initiation, as shown in previous studies (Freud and Rosenfeld, 2012; Rosenfeld and Gutman, 1994; Braga et al., 2017b). A possible explanation for the similar linear relationships for liquid and frozen precipitation is that the formation of ice particles was initiated mostly by freezing raindrops during flights AC09 and AC18, cases in which warm rain formation was not completely suppressed (Braga et al., 2017b). In addition, the assumption of an ice density of 0.9 g cm$^{-3}$ for frozen particles while calculating *PWC* can also lead to deviations in the values of the adjusted equation of $r_e$ - *PWC*. Nevertheless, similar
results were found when assuming frozen particles with lower density (0.45 g cm$^{-3}$) when calculating *PWC* (see Fig. S3).

### 3.2   Precipitation probability as a function of the measured $r_e$

Figure 3a shows the precipitation probability as a function of $r_e$ near cloud tops of convective clouds. The probability of precipitation (PP) is the fraction of in-cloud measurements (at 1 Hz) that exceed a given *DWC* threshold (e.g., for *DWC* > 0.01 g m$^{-3}$). This was calculated as a function of $r_e$ to identify the threshold of precipitation initiation. The *DWC* includes only
particles with a terminal fall speed of $\sim$ 1 m s$^{-1}$ or less, which maximizes the chance that the drizzle was formed in situ and had not fallen a large distance from above (Freud and Rosenfeld, 2012; Braga et al., 2017b). The figure shows that precipitation

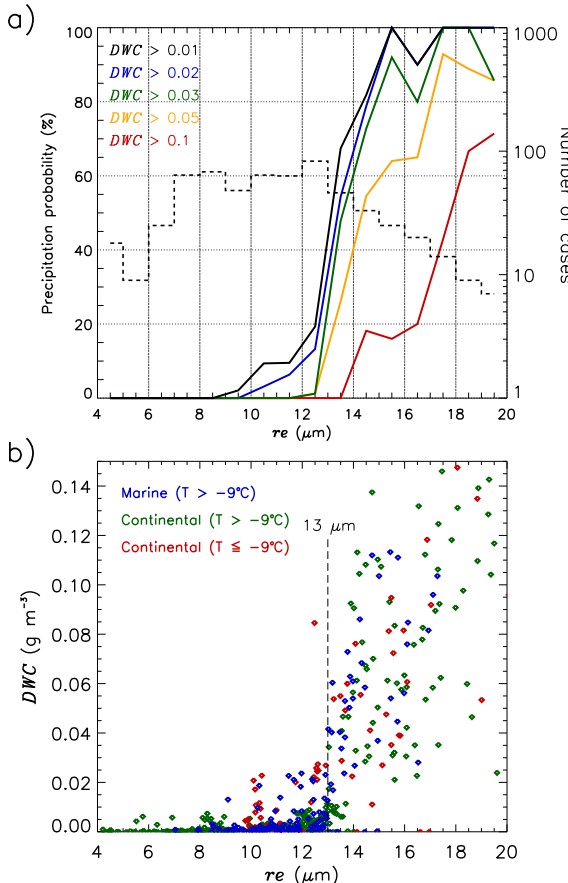

**Figure 3.** a) Precipitation probability as a function of $r_e$ for different drizzle water content (*DWC*) thresholds (black: *DWC* > 0.01 g m$^{-3}$; blue: *DWC* > 0.02 g m$^{-3}$; green: *DWC* > 0.03 g m$^{-3}$; yellow: *DWC* > 0.05 g m$^{-3}$; red: *DWC* > 0.1 g m$^{-3}$ ) measured within convective cloud tops over the Amazon Basin and Atlantic Ocean. The dashed line indicates the number of cases for each $r_e$ size interval (right axis), each case represents a 1-s in-cloud measurement. b) Effective radius ($r_e$) as a function of *DWC* measured within convective cloud tops over the Atlantic Ocean for temperatures (*T*) warmer than -9 ˚C (in blue), and over the continent for T > -9 ˚C (in green) and T ≤ -9 ˚C (in red).

initiation is expected to occur at $r_e > 13~\mu$m. It shows the greatly increased PP when $r_e$ reaches 14 $\mu$m, but some very light precipitation can occur already between 10 $\mu$m and 12 $\mu$m.

For $r_e < 13~\mu$m, a few cloud passes with light precipitation were found (see Fig. 3b). For warm temperatures, these measurements were performed over the Atlantic Ocean during flight AC19. Above the ocean, the presence of giant CCN can lead to warm rain initiation for $r_e$ below 13 $\mu$m (Freud and Rosenfeld, 2012; Konwar et al., 2012). Precipitating particles were measured for $r_e < 13~\mu$m in cloud passes with cold temperatures, in which graupel particles (probably with low density) were imaged by the CCP (see Fig. S4). This type of particles was imaged during flights AC13 and AC07, in clouds in which the coalescence process was completely suppressed, and thus precipitating particles are formed mainly by accretion. For $r_e > 13~\mu$m, $PWC$ increases rapidly as a function of $r_e$, which is probably associated with an increase of drop collision (or collision kernel) in cumulus clouds. Similar results are found for measurements with cloud water content larger than 25 % of the adiabatic water content (see Fig.S5), in which convectively diluted or dissipating clouds are excluded. This $r_e$ threshold is consistent with the result found by Braga et al. (2017b) for $r_{ec}$.

### 3.3   The relationship between $r_e$ and cloud mass

The thermal instability in the boundary layer promotes the formation of convective clouds consisting of regions with updrafts and downdrafts. Tropical convective cumuli typically develop in updrafts and during their vertical development cloud droplets are converted into precipitation by coalescence or accretion processes. Near cloud tops the amount of water or ice mass ($M$) within the cloud parcel of convective clouds can be described by:

$$M = M_c + M_p \tag{10}$$

where, $M_c$ is the mass of particles with cloud droplet sizes $1.5~\mu$m $< r < 25~\mu$m and $M_p$ is the mass of particles with precipitating sizes ($r \geq 25~\mu$m).

Figure 4a shows the particle size distribution (PSD) measured within relatively clean clouds and different resulting $r_e$ during flight AC09, while Figs. 4b and 4c show measured PSDs in marine clean clouds during flight AC19 and very polluted convective clouds during flight AC13, respectively. The figure shows that for the cleaner cases, droplets grow by coalescence to large drops and form precipitation at warmer temperatures. For the polluted case, the cloud droplets do not coalesce and the precipitation-size particles in ice phase are formed by accretion. More numerous small particles are found in polluted clouds in comparison with the clean clouds. Furthermore, for clean and polluted clouds, the number concentration of precipitating particles, and thus $M_p$, increases as a function of $r_e$.

Figure 5 shows the cloud mass ratio ($CMR$) and the precipitation mass ratio ($PMR$) in convective clouds for cloud passes with $r_e > 13~\mu$m ($\sim$ precipitation initiation threshold). This figure shows the precipitation ratio increasing as a function of $r_e$, while the cloud mass ratio is decreasing at the same time. There is a clear anti-correlation between $CMR$ and $PMR$. This inverse relationship was found because no precipitation from higher cloud regions disturbed the measurements, and thus, the formation of precipitating particles is associated with the growth of smaller particles at cloud tops.

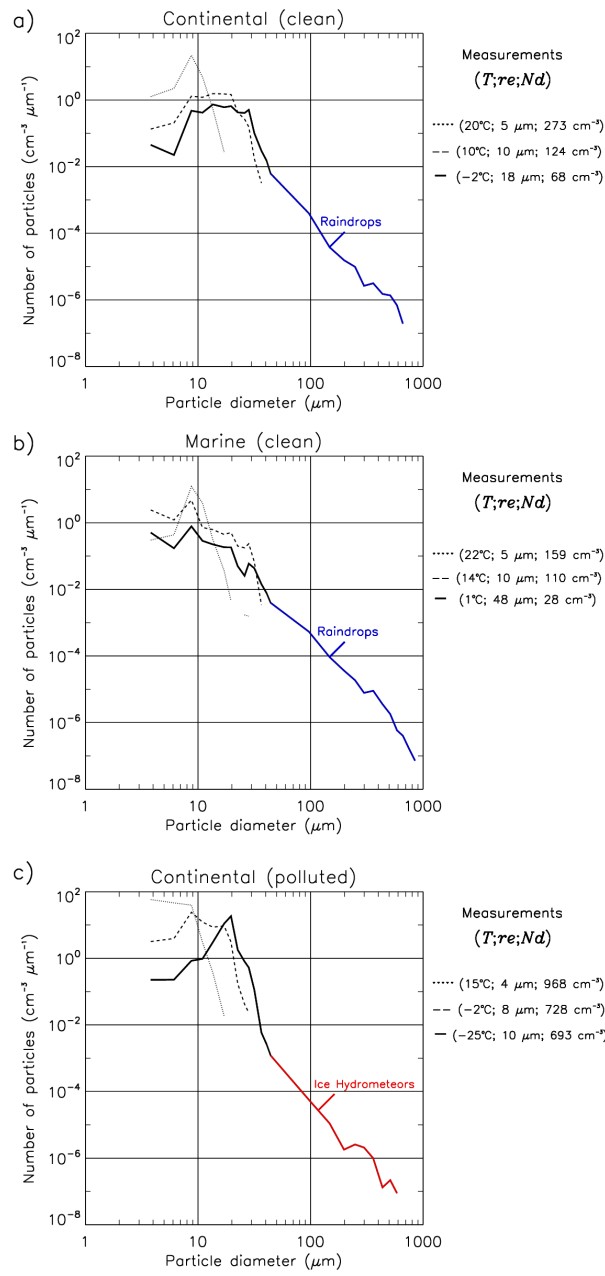

**Figure 4.** a) Number size distribution of particles in clean convective clouds measured during flight AC09 for different $r_e$ and temperatures ($T$). The values of $r_e$, the number concentration of particles ($N_d$), and $T$ for each case are shown on the right side of the panels. b) and c) are similar to a) for marine clouds measured during flight AC19 and for polluted clouds measured during flight AC13, respectively. Particles with precipitating sizes (raindrops and ice hydrometeors) are indicated by colors.

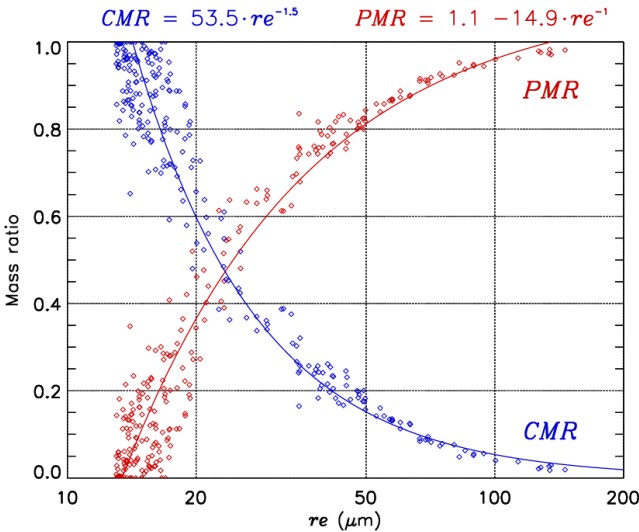

**Figure 5.** Effective radius of cloud particles ($r_e$) vs. cloud mass ratio (*CMR*) indicated by blue dots and precipitation mass ratio (*PMR*) indicated by red dots for cloud passes where $r_e > 13$ $\mu$m. The blue line indicates the best fit for measurements of *CMR* as a function of $r_e$ (the equation is indicated in blue at the top of the graph). The $R^2$ from this fit function is 0.97. The red line indicates the best fit for measurements of *PMR* as a function of $r_e$ (the equation is indicated in red at the top of the graphic). The $R^2$ from the fit function of $r_e$-*PMR* is 0.93. The number of cloud passes in this analysis is 254.

## 4   Discussion

The findings shown in this study highlight $r_e$ as a crucial quantity to define the microphysical stage of convective cloud development. We show that precipitation near the cloud tops of convective clouds can be identified and estimated with high accuracy based on in situ measurements of $r_e$. For $r_e > 13$ $\mu$m the mass of cloud drops and precipitation can be retrieved as well. Furthermore, our analysis shows that neglecting precipitating particles ($r > 25$ $\mu$m) in the calculation of $r_e$, as performed in previous studies (Freud and Rosenfeld, 2012; Braga et al., 2017b), has obscured the tight $r_e$-*PWC* relationship shown in this

study. A similar tight relationship between the size of hydrometeors and rain rates has been found initially by Marshall and Palmer (1948). These authors showed that the rain rate and hence *PWC* has a strong correlation with the raindrop diameter (*D*) and concentration. The radar reflectivity of particles depends on $D^6$ and is commonly used to estimate rain rates. Here, we show for the first time the close relationship between $r_e$ and *PWC* measured at cloud tops of convective clouds. This close relationship was found due to the inclusion of particles with precipitating sizes up to ~1 mm in diameter when deriving

the $r_e$-*PWC* relationship. Our analysis also suggests that similar linear relationships may be found for cloud particles with diameters up to 250 $\mu$m (see Fig. S6). In addition, we found a larger sensitivity of $r_e$ to precipitating particles in comparison with typical quantities used to characterize size distributions of particles (e.g., mean radius, mean volume radius, and modal

radius) (see Fig.S7). This sensitivity of $r_e$ to precipitating particles was more evident for cloud passes with precipitation formed by coalescence processes (see Fig.S8).

The relationships between in situ $r_e$ and cloud properties were found in cloud tops of growing convective cumuli. Nevertheless, similar relations between these measurements can be expected to be found in cloud tops of other types of clouds, in which the same precipitation-forming processes take place (i.e., coagulation and accretion). The applicability of our findings, e.g., for satellite measurements, depends on the sensitivity of the retrieved $r_e$ to precipitating particles at the top of clouds. Previous studies (King et al., 2013; Krisna et al., 2018; Noble and Hudson, 2015; Painemal and Zuidema, 2011) have shown good agree-

ment between in situ measurements of $r_e$ and co-located $r_e$ retrieved from the Moderate Resolution Imaging Spectroradiometer (MODIS) aboard the Aqua and Terra satellites. Furthermore, the retrieved $r_e$ from MODIS has also shown good agreement with the retrieved $r_e$ from the new generation of Chinese geostationary meteorological satellites FY-4A (Chen et al., 2020). The $r_e$ retrieved based on satellite passive infrared remote sensing represents a vertically weighted value, where the cloud top layers are weighted the most. Investigating the relationship between the $r_e$ retrieved by satellites with in situ measurements of

$PWC$ is an important task to confirm the relationship identified in the present study and to further establish $r_e$ as a parameter to quantify $PWC$ within clouds.

## 5 Conclusions

This study investigated the relationship between the effective radius, $r_e$, of droplets and ice particles and $PWC$ measured near the top of growing convective clouds. Data collected during the dry season over the Amazon Basin and over the western tropical

Atlantic with the CCP probe onboard the HALO aircraft were used in the analysis. The measurements were performed in clean and polluted air masses with cloud tops with temperatures between $\sim$ 10 ˚C and $\sim$ -26 ˚C. Our results show that rain starts at the top of most of the convective clouds over the Amazon Basin during the dry season when the measured $r_e$ exceeds about 13 $\mu$m. For marine clouds, warm rain started when $r_e$ was between 10 $\mu$m and 12 $\mu$m, probably due to the presence of giant CCN in marine air masses. In polluted air masses, in which warm rain was completely suppressed, precipitation starts at smaller $r_e$

($\sim$ 10 $\mu$m), and the observed precipitation particles were ice hydrometeors. We show for the first time that there is a clear linear relationship (R $\sim$ 0.94) between $r_e$ and $PWC$ at the tops of convective clouds. Our results also highlight that at cloud tops, the mass of cloud and precipitating particles can be estimated based on the value of $r_e$ after rain starts. These remarkable results were found because the aircraft preferably measured growing convective cloud towers near their tops, where no precipitation from higher cloud regions disturbed the in situ precipitation-forming processes. Our findings from cloud top measurements

under different thermodynamic and pollution conditions over the Amazon basin (dry and polluted air masses) and Atlantic Ocean (wet and clean air masses) suggest that the $r_e$-$PWC$ relationship can be extended for modelling and remote sensing applications. However, further analysis of the $r_e$-$PWC$ relationship including different types of precipitating clouds (at cloud tops and below), environmental conditions and others precipitating-forming processes (e.g., aggregation of ice hydrometeors) are needed to assess the limits of the applicability of this study.

*Data availability.* The data used in this study can be found at https: //halo-db.pa.op.dlr.de/mission/5.

*Competing interests.* The authors declare that they have no conflict of interest.

*Acknowledgements.* We thank the ACRIDICON-CUUVA team. The ACRIDICON-CHUVA campaign was supported by the Max Planck Society (MPG), the German Science Foundation (DFG Priority Program SPP 1294), the German Aerospace Center (DLR), and a wide range of other institutional partners. This work has also been supported by the French National Research Agency (ANR) (grant no. ANR-17-MPGA- 0013)

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
