# Peer review of "Linear relationship between effective radius and precipitation water content near the top of convective clouds: measurement results from ACRIDICON-CHUVA campaign"

_Atmospheric Chemistry and Physics, 2021_

## Referee Comment (RC1)

**Review of manuscript titled "Linear relationship between effective radius and precipitation water content near the top of convective clouds" by Ramon Campos Braga et al. submitted to EGU's *Atmospheric Chemistry and Physics by Anonymous Referee #1***

This work provides, unique aircraft measurements in clean and polluted conditions over the Amazon Basin and the western tropical Atlantic in September 2014 to come up with a threshold value of effective radius of droplets and ice particles ($r_e$) for warm rain initiation in convective clouds. This finding is consistent with previous modeling studies which indicated precipitation initiated when $r_e$ near cloud top is around 12–14 µm, as the manuscript also states in the introduction. Authors found a statistically significant linear relationship between $r_e$ and precipitation water content (PWC) with high correlation , i.e. nearly 0.94.

However, the scope of this study is limited to the 2014 dry season (September) in Amazon rain forest as stated in the manuscript. However, authors need to point towards the future research need to expand to say a wet season with different meteorological conditions and thermodynamics as well, to understand – if this linear relationship between $r_e$ and precipitation water content (PWC) holds true 'temporally' as well. If possible: add few sentences in discussion, in relation to possibility in disturbances to in-situ precipitation formation processes (See last comment #14). Also, possibility of validating in-situ measured vs satellite retrieved $r_e$ should be explored (See comment #13), if possible in reasonable time-frame for revised version of this manuscript.

The manuscript is well written with findings presented well through descriptive statistics (with various sensitivities pertaining to cloud property, size properties and pollution types) and visualizations. The findings are critical for a wider regional- and global- scale modeling community interested in correcting biases pertaining to precipitation amount in widely accepted meteorological models like WRF in WRF-Chem. I will encourage this manuscript for publication, once authors address the following edits/comments:

1) Lines 25-26 (consider rephrasing the following lines and explaining 'precipitation-forming processes' in terms of connecting them better with subsequent sentences):
   *"In the Amazon Basin, the formation and development of precipitation-forming processes of convective clouds occur at different levels of atmospheric pollution". (Also see Comment #7)*

2) Lines 31-32: Suggest defining the time-period of *'wet', 'dry' and 'dry-to-wet transition'* seasons of Amazon region, on their first use.

3) Lines 42-43: *'Amazonian dry season in September 2014'* (refer to comment #2, define time-period of different seasons at first use)

4) Lines 47-52: Please provide appropriate citations in these statements if possible:

*"Here, the relationship between cloud particle sizes and PWC is investigated by calculating retaking into account the concentration of particles with precipitating sizes (1.5μm< r ≤480μm)* **(citation needed)**. *The size range of the PWC calculation includes particles with drizzle (25μm≤ r ≤125μm) and raindrop (125μm< r ≤480μm) sizes* **(citation needed)**. *This size range is selected because it includes particles with terminal fall50speeds large enough ( >~0.5 m s⁻¹) to survive evaporative dissipation over a distance of the order of several hundred meters* **(citation needed)**. *Droplets smaller than drizzle particles fall slowly enough from most clouds that they evaporate before reaching the ground* **(citation needed)**.*"*

5) Lines 53-54: Possibly would be better to rephrase *'……which increases with the 5ᵗʰ power of $r_{ec}$'* with something on the lines of:
   *'…..Coalescence rate increases in direct proportion to $r_{ec}^5$'*?

6) Lines 57-58: Please add citation if possible: *'For raindrops, this value is close to unity, and is several times larger than that for small drops (r< 10μm).'* OR
   Better to combine following 2 sentences if they are from same citation:
   *'The collision efficiency of drops increases as a function of their sizes (Khain and Pinsky, 2018). For raindrops, this value is close to unity, and is several times larger than that for small drops (r< 10μm).'*

7) Line 59: Why 'graupel' form of ice drops are mentioned specifically? Are other types such as hail or sleet uncommon? If so why? Also, would be helpful to briefly mention at first instance of use of what 'graupel' ice form means physically or size-wise (and its difference with 'frozen' form), if possible. I notice some features of graupel ice drops is mentioned in Line 134, but introduction early on is more apt.

8) Lines 62-63: *'…..These **precipitation-forming processes** result in a broadening of the particle size distribution and thus $r_e$.'* (See comment #1, where *'precipitation-forming processes'* are mentioned first in Lines 25-26 but explained much later here). Some rearrangement of text would be better to have these parts next to each other sequentially.

9) General comment for Abstract and Introduction:

   *How the term 'Amazon Basin and over the western tropical Atlantic'* is mentioned in both "Abstract" and "Introduction" sections might confuse readers who are not very familiar with HALO or ACRIDICON-CHUVA flight campaign, as though the manuscript will present data for 2 separate flight campaigns one over Amazon and other over Atlantic. Would be better to clarify that the HALO flights cover this entire region as a single campaign, at the first instance of use in introduction/abstract. (It is clear in Figure 1 of course)

**10)** Figure 1 and Lines 77-80: As the Flights are color-coded as per pollution level classification in Figure 1, might help to add that in Figure 1 caption text and in the following preceding text as well for ease of readers:
*'Convective clouds formed in clean air masses were found above the Atlantic Ocean during flight AC19* **(in blue, Fig. 1)**. *Flights AC09 and AC18 took place in lightly polluted conditions* **(in green, Fig. 1)** *over the tropical rain forest. Clouds forming in deforested regions in very polluted (biomass burning) environments were measured during flights AC07 and AC13* **(in red, Fig. 1)**.*'* (Like it's done in Figure 2 caption text)

**11)** Line 84: Full forms of CDP and CIPgs in 'CCP-**CDP** and CCP–**CIPgs'** if possible should be defined – since it is first use of these abbreviations

**12)** Lines 87-89: Can you back up the following criteria for cloud pass with a suitable citation or elaborate on it further? :

*'In this study, a cloud pass is assumed when the total water content (TWC) exceeds 0.05 g $m^{-3}$ and the number concentration of drops ($N_d$) exceeds 20 $cm^{-3}$. This is performed to avoid cloud passes well mixed with environment air.'*

**13)** Lines 2019-217: Is comparison of co-located $r_e$ from MODIS satellite retrievals with $r_e$ presented as part of the measurements in this manuscript, more of a future research step or plausible to be included to validate this linear relationship between $r_e$ and PWC better?

**14)** Lines 228-230: Since the author mentioned themselves: *'These remarkable results were found because at the cloud tops, no precipitation from higher cloud regions disturbed the in situ precipitation-forming processes.'*

Can authors clarify further, If its mostly because their study was in dry season and this issue of higher cloud regions with more significant rain disturbing the in-situ precipitation formation processes would perturb the smooth linear relationship between $r_e$ and PWC (or CMR, PMR as in Fig. 5), that we observe in this manuscript?

---

## Referee Comment (RC3)

In this study, the authors investigate the relationship between the effective radius of droplets and ice particles and the precipitation water content at the top of convective clouds. They use flight measurements with different pollution levels over the Amazon Basin from the ACRIDICON-CHUVA campaign in September 2014.

I thank the authors for this nice article and recommend it for publication after minor revisions. Please find my comments below.

- Title: I think the title is too general. The linear relation was only found for the flight measurements in September 2014. Maybe try to rephrase the title and include e.g. the campaign name. For example 'Linear relationship between effective radius and precipitation water content near the top of convective clouds- In situ measurements from the ACRIDICON-CHUVA campaign in 2014'

- Abstract: The campaign name ACRIDICON-CHUVA should be also stated in the abstract.

- Lines 32-35: Please add more recent references describing the Amazonian dry season.

- Extent the introduction a little bit with a more general description and references to aerosol-cloud interactions and the study domain.

- Lines 45, 50: Please add references to the chosen size ranges.

- Lines 46-63: This would fit better in the Methods section.

- Fig. 1: Please state the color for the different flights also in the figure caption.

- Lines 85, 86: A short description , just one or two sentences, about the CCP would be good.

- Lines 95-102: Please revise this part. Lines 98,99: The precipitation probability is also described in Section 3.1. Lines 100,101: This should be Section 3.2. There is no Section 3.3. as stated in your text. But I think splitting up the Results section in 3.1, 3.2 and 3.3. would be good.

- Equations 3)-9): You should refer to the equation number in the results section.

- Line 137: How small is the uncertainty in $r_e$ and PWC?

- Results section: In general, more referencing to the different flights would be interesting.

- Discussion and Conclusions sections: It should be emphasized here that the linear relationship was only found for the campaign measurements in 2014 in the Amazonian dry-season.

---

## Author Comment (AC1)

**Author response to referee comments**

**Referee 1**

We thank the referee for the detailed comments and the good suggestions for improving the paper. We have addressed all comments as listed below which improved our manuscript. Referee comments are in black, our responses in blue and manuscript text in *italic* and new text in red.

**Review of manuscript titled "Linear relationship between effective radius and precipitation water content near the top of convective clouds" by Ramon Campos Braga et al. submitted to EGU's Atmospheric Chemistry and Physics by Anonymous Referee #1**

This work provides, unique aircraft measurements in clean and polluted conditions over the Amazon Basin and the western tropical Atlantic in September 2014 to come up with a threshold value of effective radius of droplets and ice particles (re) for warm rain initiation in convective clouds. This finding is consistent with previous modeling studies which indicated precipitation initiated when re near cloud top is around 12–14 µm, as the manuscript also states in the introduction. Authors found a statistically significant linear relationship between re and precipitation water content (PWC) with high correlation, i.e. nearly 0.94.

However, the scope of this study is limited to the 2014 dry season (September) in Amazon rain forest as stated in the manuscript. However, authors need to point towards the future research need to expand to say a wet season with different meteorological conditions and thermodynamics as well, to understand – if this linear relationship between re and precipitation water content (PWC) holds true 'temporally' as well. If possible: add few sentences in discussion, in relation to possibility in disturbances to in-situ precipitation formation processes (See last comment #14). Also, possibility of validating in-situ measured vs satellite retrieved re should be explored (See comment #13), if possible in reasonable time-frame for revised version of this manuscript.

The manuscript is well written with findings presented well through descriptive statistics (with various sensitivities pertaining to cloud property, size properties and pollution types) and visualizations. The findings are critical for a wider regional- and global- scale modeling community interested in correcting biases pertaining to precipitation amount in widely accepted meteorological models like WRF in WRF-Chem. I will encourage this manuscript for publication, once authors address the following edits/comments:

1) Lines 25-26 (consider rephrasing the following lines and explaining 'precipitation-forming processes' in terms of connecting them better with subsequent sentences): "In the Amazon Basin, the formation and development of precipitation-forming processes of convective clouds occur at different levels of atmospheric pollution". (Also see Comment #8)

Author responses: In order to address this comment and additional comments regarding the Introduction section we have re-written the paragraphs #3 and #4 of this section. The new text is below:

*"Braga et al., (2017b) have described the general characteristics of growing convective cumulus formed over the Amazon basin and Atlantic Ocean based on in situ measurements. The*

*measurements were performed during the ACRIDICON-CHUVA (Aerosol, Cloud, Precipitation, and Radiation Interactions and Dynamics of Convective Cloud Systems–Cloud Processes of the Main Precipitation Systems in Brazil: A Contribution to Cloud Resolving Modeling and to the Global Precipitation measurements) campaign in the Amazonian dry season in September 2014 (Wendisch et al., 2016). During the campaign, cloud profiling flights were performed in regions of different pollution levels and thermodynamic conditions. Braga et al., (2017b) showed that the heights of cloud base are higher over the continental Amazon due to the smaller relative humidity in comparison with the maritime region. Convective clouds formed over the Atlantic Ocean near the Brazilian coast have smaller cloud droplet concentrations at cloud base due to the smaller concentration of aerosol and updraft velocities below cloud base. For convective clouds formed over forested and deforested regions, larger aerosol concentration and updrafts were observed below cloud base, leading to larger droplet concentrations activated at cloud base. The precipitating particles were formed mostly by coalescence of drops at temperatures above 0˚C over ocean. Over the forest, lightly polluted air masses were found and the precipitation initiation (liquid raindrops) was observed near 0˚C. For very polluted air masses, found over the deforestation arc region, the collision and coalescence processes were totally suppressed and the formation of precipitating particles took place at higher altitudes as ice hydrometeors. In these cases, precipitating particles were formed mostly by accretion processes at temperatures below 0˚C, when the growth of ice hydrometeors took place from collision with supercooled drops that freeze completely or partially upon contact.*

*The relationship between particle sizes and precipitation is associated with the coalescence rate which increases in direct proportion to the cloud droplet effective radius ($r_{ec}^5$) (Freud and Rosenfeld, 2012). Previous studies have found $r_{ec}$ between 13 µm and 14 µm as a suitable threshold for precipitation initiation (Freud and Rosenfeld, 2012; Rosenfeld and Gutman, 1994; Braga et al., 2017b). The relation between rain initiation and $r_{ec}$ is associated with the increase of both the drop-swept volume and collision efficiency. The collision efficiency of drops increases as a function of their sizes (Khain and Pinsky, 2018). For raindrops, this value is close to unity, and is several times larger than that for small drops (r < 10 µm) (Pinsky and Khain, 2002). The collision and coalescence processes of liquid drops and the accretion processes at supercooled temperatures have strong effect on the broadening of the particle size distribution and thus particle sizes. In this study, we have investigated measurements of the effective radius ($r_e$) of cloud particles and the rain and ice precipitation water content (PWC) using data from cloud probes, measured at the cloud tops of growing convective cumulus during the ACRIDICON-CHUVA campaign. We focus our analysis on flights in which precipitation was found in the cloud tops of convective clouds. Our findings shown in the next sections describe the tight relationship between $r_e$ and PWC for in situ measurements in cloud tops in different pollution states and temperature levels. We show that $r_e$ determines both the initiation and amount of precipitation at the top of convective clouds."*

      2) Lines 31-32: Suggest defining the time-period of 'wet', 'dry' and 'dry-to-wet transition' seasons of Amazon region, on their first use.

      3) Lines 42-43: 'Amazonian dry season in September 2014' (refer to comment #2, define time-period of different seasons at first use)

Author responses to #2 and #3: We have defined: wet season (Feb-May) and dry season (Aug-Oct). Thanks.

4) Lines 47-52: Please provide appropriate citations in these statements if possible:"Here, the relationship between cloud particle sizes and PWC is investigated by calculating retaking into account the concentration of particles with precipitating sizes (1.5µm< r ≤480µm) (citation needed). The size range of the PWC calculation includes particles with drizzle (25µm≤ r ≤125µm) and raindrop (125µm< r ≤480µm) sizes (citation needed). This size range is selected because it includes particles with terminal fall speeds large enough ( >∼0.5 m s-1) to survive evaporative dissipation over a distance of the order of several hundred meters (citation needed). Droplets smaller than drizzle particles fall slowly enough from most clouds that they evaporate before reaching the ground (citation needed)."

Author responses: We have provided these references. This text is now on Methods section as suggested by referee #2. Below is the new text:

"Previous studies (e.g., Freud and Rosenfeld, 2012; Braga et al., 2017b) have calculated $r_e$ using data of particle number concentration with radii between 1.5 µm and 25 µm ($r_{ec}$), which does not include precipitating particles. Here, the relationship between cloud particle sizes and PWC is investigated by calculating $r_e$ taking into account the concentration of all the measured particles including precipitating sizes (1.5 µm < r ≤ 480 µm). Precipitating particles are considered those with terminal fall speeds large enough (> ∼ 0.5 m s$^{-1}$) to survive evaporative dissipation over a distance of the order of several hundred meters (Beard, 1976). Droplets smaller than drizzle particles fall slowly enough from most clouds that they evaporate before reaching the ground. The size range of the PWC calculation includes particles with drizzle (25 µm ≤ r ≤ 125 µm) and raindrop (125 µm < r ≤ 480 µm) sizes. The drizzle water content (DWC) and PWC are calculated using the size range of drizzle and raindrop in Eq. 2. Precipitating particles in this size range (r > 25 µm) were often imaged by cloud probes within convective cumulus during ACRIDICON-CHUVA campaign (Braga et al., 2017b)."

References:

Beard, K.V., 1976: Terminal velocity and shape of cloud and precipitation drops aloft. J. Atmos. Sci., 33, 851–864.

Braga, R. C., Rosenfeld, D., Weigel, R., Jurkat, T., Andreae, M. O., Wendisch, M., Pöschl, U., Voigt, C., Mahnke, C., Borrmann, S., Albrecht, R. I., Molleker, S., Vila, D. A., Machado, L. A. T., and Grulich, L.: Further evidence for CCN aerosol concentrations determin255 ing the height of warm rain and ice initiation in convective clouds over the Amazon basin, Atmos. Chem. Phys., 17, 14 433–14 456,https://doi.org/10.5194/acp-17-14433-2017, 2017b.

5) Lines 53-54: Possibly would be better to rephrase '……which increases with the 5th power of rec' with something on the lines of: '…..Coalescence rate increases in direct proportion to rec5 '?

Author responses: OK. We have changed as you suggested. Thanks

6) Lines 57-58: Please add citation if possible: 'For raindrops, this value is close to unity, and is several times larger than that for small drops (r< 10µm).' OR Better to combine following 2 sentences if they are from same citation: 'The collision efficiency of drops increases as a function of their sizes (Khain and Pinsky, 2018). For raindrops, this value is close to unity, and is several times larger than that for small drops (r< 10µm).'

Author responses: We added the citation for the sentence as follow:

"For raindrops, this value is close to unity, and is several times larger than that for small drops (r < 10 µm) (Pinsky and Khain, 2002)."

*Reference:*

*Pinsky, M. B., and A. P. Khain. "Effects of in-cloud nucleation and turbulence on droplet spectrum formation in cumulus clouds." Quarterly Journal of the Royal Meteorological Society: A journal of the atmospheric sciences, applied meteorology and physical oceanography 128.580 (2002): 501-533.*

7) Line 59: Why 'graupel' form of ice drops are mentioned specifically? Are other types such as hail or sleet uncommon? If so why? Also, would be helpful to briefly mention at first instance of use of what 'graupel' ice form means physically or size-wise (and its difference with 'frozen' form), if possible. I notice some features of graupel ice drops is mentioned in Line 134, but introduction early on is more apt.

Author responses: We have mentioned in line 128 that we found mostly graupel and frozen drops because it was imaged by the probe CCP-CIP (see images at Braga et al., 2017b). Other types of ice particles were imaged by CIP but with smaller frequency. Results assuming smaller density for ice particles are also shown in the supplement.

8) Lines 62-63: '…..These precipitation-forming processes result in a broadening of the particle size distribution and thus re.' (See comment #1, where 'precipitation-forming processes' are mentioned first in Lines 25-26 but explained much later here). Some rearrangement of text would be better to have these parts next to each other sequentially.

Author responses: We have rearranged the text as suggested by the referee (see the response for comment #1).

9) General comment for Abstract and Introduction: How the term 'Amazon Basin and over the western tropical Atlantic' is mentioned in both "Abstract" and "Introduction" sections might confuse readers who are not very familiar with HALO or ACRIDICON-CHUVA flight campaign, as though the manuscript will present data for 2 separate flight campaigns one over Amazon and other over Atlantic. Would be better to clarify that the HALO flights cover this entire region as a single campaign, at the first instance of use in introduction/abstract. (It is clear in Figure 1 of course)

Author responses: OK. We have re-written paragraphs #3 and #4 at Introduction section to address these comments (see response for comment #1). We have also rephrased the abstract as follow:

*"The data for this study were collected during the ACRIDICON-CHUVA campaign on the HALO research aircraft in clean and polluted conditions over the Amazon Basin and over the western tropical Atlantic in September 2014."*

10) Figure 1 and Lines 77-80: As the Flights are color-coded as per pollution level classification in Figure 1, might help to add that in Figure 1 caption text and in the following preceding text as well for ease of readers: 'Convective clouds formed in clean air masses were found above the Atlantic Ocean during flight AC19 (in blue, Fig. 1). Flights AC09 and AC18 took place in lightly polluted conditions (in green, Fig. 1) over the tropical rain forest. Clouds forming in deforested regions in very polluted (biomass burning) environments were measured during flights AC07 and AC13 (in red, Fig. 1).' (Like it's done in Figure 2 caption text)

Author responses: OK. We added this information in the text and Figure 1 captions. Thanks.

11) Line 84: Full forms of CDP and CIPgs in 'CCP-CDP and CCP–CIPgs' if possible should be defined – since it is first use of these abbreviations

Author responses: OK. We added this information and additional description of CCP in the new version as follow:

*"The CCP combines two detectors, the Cloud Droplet Probe (CDP) and the grayscale Cloud Imaging Probe (CIPgs). The CDP is an open-path instrument that detects forward-scattered laser light from cloud particles as they pass through the detection area (Lance et al., 2010). The CIP records 2-D shadow-cast images of cloud elements. The identification of water drops and ice hydrometeors were performed by Braga et al., (2017b) from the occurrence of visually spherical and non-spherical shapes of the shadows. The combination of CCP–CDP and CCP–CIPgs information provides the ability to measure particles within clouds for nearly the same air sample volume."*

12) Lines 87-89: Can you back up the following criteria for cloud pass with a suitable citation or elaborate on it further? : 'In this study, a cloud pass is assumed when the total water content (TWC) exceeds 0.05 g m-3 and the number concentration of drops (Nd) exceeds 20 cm-3. This is performed to avoid cloud passes well mixed with environment air.'

Author responses: Sure. We added new sentences including a reference as follow:

*"In this study, a cloud pass is assumed when the total water content (TWC) exceeds 0.05 g m$^{-3}$ and the number concentration of drops ($N_d$) exceeds 20 cm$^{-3}$. This criterion was applied to avoid cloud passes well mixed with subsaturated environment air (RH < 100%) and counts of haze particles, typically found at cloud edges and dissipating convective clouds during the ACRIDICON-CHUVA campaign (Braga et al.,2017a)."*

13) Lines 2019-217: Is comparison of co-located re from MODIS satellite retrievals with re presented as part of the measurements in this manuscript, more of a future research step or plausible to be included to validate this linear relationship between re and PWC better?

Author responses: The comparison between satellite $r_e$ and in situ measurements of $r_e$ and PWC at cloud top of convective clouds is a future step. The limited overlap between satellite overpass and aircraft data do not provide enough statistic for such comparison in the scope of this study.

14) Lines 228-230: Since the author mentioned themselves: 'These remarkable results were found because at the cloud tops, no precipitation from higher cloud regions disturbed the in situ precipitation-forming processes.' Can authors clarify further, If its mostly because their study was in dry season and this issue of higher cloud regions with more significant rain disturbing the in-situ precipitation formation processes would perturb the smooth linear relationship between re and PWC (or CMR, PMR as in Fig. 5), that we observe in this manuscript?

Author responses: we have rephrased the conclusions to address this comment as follow:

*"These remarkable results were found because the aircraft preferably measured growing convective cloud towers near their tops, where no precipitation from higher cloud regions disturbed the in situ precipitation-forming processes. Our findings from cloud top measurements under different thermodynamic and pollution conditions over the Amazon basin (dry and polluted air masses) and Atlantic Ocean (wet and clean air masses) suggest that the $r_e$-PWC relationship can be extended for modelling and remote sensing applications. However, further analysis of the $r_e$-PWC relationship including different types of precipitating clouds (below and at cloud tops), environmental conditions and precipitating-forming processes (e.g., aggregation of ice hydrometeors) are needed to assess the limits of the applicability of the results of this study"*

**Author response to referee comments**

Referee 2

We thank the referee for the detailed comments and the good suggestions for improving the paper. We have addressed all comments as listed below which improved our manuscript. Referee comments are in black, our responses in blue and manuscript text in *italic* and new text in *red*.

**Review of manuscript titled "Linear relationship between effective radius and precipitation water content near the top of convective clouds" by Ramon Campos Braga et al. submitted to EGU's Atmospheric Chemistry and Physics by Anonymous Referee #2**

In this study, the authors investigate the relationship between the effective radius of droplets and ice particles and the precipitation water content at the top of convective clouds. They use flight measurements with different pollution levels over the Amazon Basin from the ACRIDICON-CHUVA campaign in September 2014.

I thank the authors for this nice article and recommend it for publication after minor revisions. Please find my comments below.

• Title: I think the title is too general. The linear relation was only found for the flight measurements in September 2014. Maybe try to rephrase the title and include e.g. the campaign name. For example 'Linear relationship between effective radius and precipitation water content near the top of convective clouds- In situ measurements from the ACRIDICON-CHUVA campaign in 2014'

Author responses: we have changed the title to address this suggestion as follow:

"Linear relationship between effective radius and precipitation water content near the top of convective clouds: measurement results from ACRIDICON-CHUVA campaign"

• Abstract: The campaign name ACRIDICON-CHUVA should be also stated in the abstract.

Author responses: we have included the campaign name in the abstract. Thanks.

• Lines 32-35: Please add more recent references describing the Amazonian dry season.

Author responses: we have added the following references:

Artaxo, P.: Physical and chemical properties of aerosols in the wet and dry seasons in Rondônia, Amazonia, Journal of Geophysical Research,107, 8081, https://doi.org/10.1029/2001JD000666, 2002.

Artaxo, P., Rizzo, L.V., Brito, J.F., Barbosa, H.M.J., Arana, A., Sena, E.T., Cirino, G.G., Bastos, W., Martin, S.T., Andreae, M.O., 2013. Atmospheric aerosols in Amazonia and land use change: from natural biogenic to biomass burning conditions. Faraday Discuss. 165, 203. http://dx.doi.org/10.1039/c3fd00052d.

Pöhlker, M. L., Ditas, F., Saturno, J., Klimach, T., Hrabˇe De Angelis, I., Araùjo, A. C., Brito, J., Carbone, S., Cheng, Y., Chi, X., Ditz, R.,Gunthe, S. S., Holanda, B. A., Kandler, K., Kesselmeier, J., Könemann, T., Krüger, O. O., Lavric, J. V., Martin, S. T., Mikhailov, E.,Moran-Zuloaga, D., Rizzo, L. V., Rose, D., Su, H., Thalman, R., Walter, D., Wang, J., Wolff, S., Barbosa, H. M., Artaxo, P., Andreae,M. O., Pöschl, U., and Pöhlker, C.: Long-term observations of cloud condensation nuclei over the Amazon rain forest - Part 2: Variability and characteristics of biomass burning, long-range transport, and pristine rain forest aerosols, Atmospheric Chemistry and Physics,https://doi.org/10.5194/acp-18-10289-2018, 2018.

Roberts, G. C., Nenes, A., Seinfeld, J. H., and Andreae, M. O.: Impact of biomass burning on cloud properties in the Amazon Basin - art. no.4062, Journal of Geophysical Research - Atmospheres, 108, 4062, https://doi.org/10.1029/2001JD000985, 2003.

       • Extent the introduction a little bit with a more general description and references to aerosol-cloud interactions and the study domain.

Author responses: we have added the following paragraphs in the introduction. The paragraphs #3 and #4 from introduction were re-written.

*"Braga et al., (2017b) have described the general characteristics of growing convective cumulus formed over the Amazon basin and Atlantic Ocean based on in situ measurements. The measurements were performed during the ACRIDICON-CHUVA (Aerosol, Cloud, Precipitation, and Radiation Interactions and Dynamics of Convective Cloud Systems–Cloud Processes of the Main Precipitation Systems in Brazil: A Contribution to Cloud Resolving Modeling and to the Global Precipitation measurements) campaign in the Amazonian dry season in September 2014 (Wendisch et al., 2016). During the campaign, cloud profiling flights were performed in regions of different pollution levels and thermodynamic conditions. Braga et al., (2017b) showed that the heights of cloud base are higher over the continental Amazon due to the smaller relative humidity in comparison with the maritime region. Convective clouds formed over the Atlantic Ocean near the Brazilian coast have smaller cloud droplet concentrations at cloud base due to the smaller concentration of aerosol and updraft velocities below cloud base. For convective clouds formed over forested and deforested regions, larger aerosol concentration and updrafts were observed below cloud base, leading to larger droplet concentrations activated at cloud base. The precipitating particles were formed mostly by coalescence of drops at temperatures above 0˚C over ocean. Over the forest, lightly polluted air masses were found and the precipitation initiation (liquid raindrops) was observed near 0˚C. For very polluted air masses, found over the deforestation arc region, the collision and coalescence processes were totally suppressed and the formation of precipitating particles took place at higher altitudes as ice hydrometeors. In these cases, precipitating particles were formed mostly by accretion processes at temperatures below 0˚C, when the growth of ice hydrometeors took place from collision with supercooled drops that freeze completely or partially upon contact.*

*The relationship between particle sizes and precipitation is associated with the coalescence rate which increases in direct proportion to the cloud droplet effective radius ($r_{ec}^5$) (Freud and Rosenfeld, 2012). Previous studies have found $r_{ec}$ between 13 μm and 14 μm as a suitable threshold for precipitation initiation (Freud and Rosenfeld, 2012; Rosenfeld and Gutman, 1994; Braga et al., 2017b). The relation between rain initiation and $r_{ec}$ is associated with the increase of both the drop-swept volume and collision efficiency. The collision efficiency of drops increases as a function of their sizes (Khain and Pinskˇy, 2018). For raindrops, this value is close*

*to unity, and is several times larger than that for small drops (r < 10 μm) (Pinsky and Khain, 2002). The collision and coalescence processes of liquid drops and the accretion processes at supercooled temperatures have strong effect on the broadening of the particle size distribution and thus particle sizes. In this study, we have investigated measurements of the effective radius ($r_e$) of cloud particles and the rain and ice precipitation water content (PWC) using data from cloud probes, measured at the cloud tops of growing convective cumulus during the ACRIDICON-CHUVA campaign. We focus our analysis on flights in which precipitation was found in the cloud tops of convective clouds. Our findings shown in the next sections describe the tight relationship between $r_e$ and PWC for in situ measurements in cloud tops in different pollution states and temperature levels. We show that $r_e$ determines both the initiation and amount of precipitation at the top of convective clouds."*

• Lines 45, 50: Please add references to the chosen size ranges.

Author responses: we have added the references.

• Lines 46-63: This would fit better in the Methods section.

Author responses: Ok. We have changed part of this paragraph to section 2.2. The paragraphs #3 and #4 from introduction were re-written.

• Fig. 1: Please state the color for the different flights also in the figure caption.

Author responses: Done. Thanks.

• Lines 85, 86: A short description , just one or two sentences, about the CCP would be good.

Author responses: we have added the following information.

"The CCP combines two detectors, the Cloud Droplet Probe (CDP) and the grayscale Cloud Imaging Probe (CIPgs). The CDP is an open-path instrument that detects forward-scattered laser light from cloud particles as they pass through the detection area Lance et al., (2010). The CIP records 2-D shadow-cast images of cloud elements. The identification of water drops and ice hydrometeors were performed by Braga et al., (2017b) from the occurrence of visually spherical and non-spherical shapes of the shadows. The combination of CCP–CDP and CCP–CIPgs information provides the ability to measure particles within clouds for nearly the same air sample volume."

• Lines 95-102: Please revise this part. Lines 98,99: The precipitation probability is also described in Section 3.1. Lines 100,101: This should be Section 3.2. There is no Section 3.3. as stated in your text. But I think splitting up the Results section in 3.1, 3.2 and 3.3. would be good.

Author responses: Many Thanks…We have splitting up the Results sections in 3.1, 3.2 and 3.3.

• Equations 3)-9): You should refer to the equation number in the results section.

• Line 137: How small is the uncertainty in re and PWC?

Author responses: We have stated the uncertainties as follow:

*"Therefore, based on the type of particles imaged by the CIPgs during our measurements we assume that the uncertainty in the calculated $r_e$ and PWC is small in comparison to the measurement uncertainty (i.e., ~10% and ~30% for $r_e$ and PWC, respectively)."*

• Results section: In general, more referencing to the different flights would

be interesting.

Author responses: Ok. We have explained better the regions of study (ocean, forested and deforestation arc) in which the flights took place in the introduction. The new paragraphs #3 and #4 improve the description of the flights shown at Figure 1. Furthermore, we added better captions at figure 1.

• Discussion and Conclusions sections: It should be emphasized here that

the linear relationship was only found for the campaign measurements in

2014 in the Amazonian dry-season.

Author responses: We have stated the following sentences in the conclusions:

*"Our findings from cloud top measurements under different thermodynamic and pollution conditions over the Amazon basin (dry and polluted air masses) and Atlantic Ocean (wet and clean air masses) suggest that the $r_e$-PWC relationship can be extended for modelling and remote sensing applications. However, further analysis of the $r_e$-PWC relationship including different types of precipitating clouds (below and at cloud tops), environmental conditions and precipitating-forming processes (e.g., aggregation of ice hydrometeors) are needed to assess the limits of the applicability of the results of this study."*